# Potential Therapeutic Targets for Combination Antibody Therapy against *Pseudomonas aeruginosa* Infections

**DOI:** 10.3390/antibiotics10121530

**Published:** 2021-12-14

**Authors:** Luke L. Proctor, Whitney L. Ward, Conner S. Roggy, Alexandra G. Koontz, Katie M. Clark, Alyssa P. Quinn, Meredith Schroeder, Amanda E. Brooks, James M. Small, Francina D. Towne, Benjamin D. Brooks

**Affiliations:** 1Department of Biomedical Sciences, College of Osteopathic Medicine, Rocky Vista University, Parker, CO 80134, USA; luke.proctor@rvu.edu (L.L.P.); whitney.ward@rvu.edu (W.L.W.); Conner.roggy@rvu.edu (C.S.R.); alexandra.koontz@rvu.edu (A.G.K.); katie.clark@rvu.edu (K.M.C.); Alyssa.Palmer@rvu.edu (A.P.Q.); abrooks@rvu.edu (A.E.B.); jsmall@rvu.edu (J.M.S.); 2Inovan, Inc., Fargo, ND 58102, USA; mschroeder@inovaninc.com

**Keywords:** polyclonal antibodies, antibiotic resistance, antibiotics, combination therapies, *Pseudomonas aeruginosa*, immunotherapies

## Abstract

Despite advances in antimicrobial therapy and even the advent of some effective vaccines, *Pseudomonas aeruginosa (P. aeruginosa)* remains a significant cause of infectious disease, primarily due to antibiotic resistance. Although *P. aeruginosa* is commonly treatable with readily available therapeutics, these therapies are not always efficacious, particularly for certain classes of patients (e.g., cystic fibrosis (CF)) and for drug-resistant strains. Multi-drug resistant *P. aeruginosa* infections are listed on both the CDC’s and WHO’s list of serious worldwide threats. This increasing emergence of drug resistance and prevalence of *P. aeruginosa* highlights the need to identify new therapeutic strategies. Combinations of monoclonal antibodies against different targets and epitopes have demonstrated synergistic efficacy with each other as well as in combination with antimicrobial agents typically used to treat these infections. Such a strategy has reduced the ability of infectious agents to develop resistance. This manuscript details the development of potential therapeutic targets for polyclonal antibody therapies to combat the emergence of multidrug-resistant *P. aeruginosa* infections. In particular, potential drug targets for combinational immunotherapy against *P. aeruginosa* are identified to combat current and future drug resistance.

## 1. Introduction

*P. aeruginosa* is a Gram-negative bacillus implicated in a wide variety of human infections. In acute infections, individual *P. aeruginosa* organisms exhibit swarming motility via a single flagellum and type 4 pili and express a wide variety of toxins, cell surface proteins, and other molecules that contribute to its immunogenicity and pathogenicity [1]. In order to establish chronic infection, *P. aeruginosa* transitions to a sessile, non-motile state marked by the formation of a mucoid biofilm, composed mainly of exo-polysaccharides, glycolipids, and mucin, which often poses a barrier to successful clinical treatment [2]. Regardless of if *P. aeruginosa* exists in an acute motile form or a chronic sessile biofilm, infection with *P. aeruginosa* is particularly perilous for immunosuppressed patients [1], ventilator-dependent patients, and cystic fibrosis patients. According to the CDC, *P. aeruginosa* infections were responsible for 32,600 nosocomial infections and 2700 deaths in 2017. Data collected from over 4500 hospitals in the United States’ National Healthcare Safety Network from 2011 to 2014 revealed the following rates of multidrug resistance among *P. aeruginosa* isolates [3]:

Ventilator-associated pneumonia—20%

Central line-associated bloodstream infection—18%

Catheter-associated urinary tract infection—18%

Surgical site infection—4%

This culminates in an estimated cost to the healthcare system of USD 767 million [4]. In cystic fibrosis patients alone, mean healthcare costs per patient increase by 87% once a patient becomes colonized with *P. aeruginosa,* to nearly USD 67,000 annually [4]. Additionally, *P. aeruginosa* has been recognized as the causative organism in catheter-associated urinary tract infections, otitis externa, otitis media, contact lens keratitis, soft tissue infections in burn victims and AIDS patients, septic arthritis, folliculitis, meningitis, and sepsis. In fact, this broad array of associated disease states (Figure 1) has led *P. aeruginosa* to be recognized as the sixth leading cause of hospital-acquired infections, the second most common cause of ventilator-associated pneumonia and the most common multidrug-resistant Gram-negative cause of ventilator-associated pneumonia, the third most common cause of catheter-associated UTI, and the fifth most common cause of surgical site infections [1].

The vast array of infectious complications that can arise from normal commensal and environmental strains of *P. aeruginosa* indicates that it is an opportunistic, adaptable, common environmental pathogen, making *P. aeruginosa* very robust and difficult to treat. Several antimicrobial agents possess the ability to treat *P. aeruginosa* infections [3]; however, successful clinical treatment regimens should include pre-treatment sensitivity testing, as different strains possess widely different antimicrobial resistances. Importantly, treatment is often dictated by the antibiogram of a specific hospital or region. *P. aeruginosa* is often susceptible to first-line agents, including beta-lactam antibiotics (e.g., piperacillin-tazobactam and ticarcillin-clavulanate), cephalosporins (e.g., ceftazidime, cefoperazone, and cefepime), and monobactams (e.g., Aztreonam). Carbapenems (e.g., meropenem and doripenem), which were historically seen as the “big gun”, last-ditch antimicrobials, can be used to treat highly resistant infections. However, as of 2019, the World Health Organization has listed carbapenem-resistant *P. aeruginosa* as one of three bacterial diseases in critical need of new treatment strategies, with up to 14% of *P. aeruginosa* isolates in the U.S. in 2019 expressing carbapenem resistance (Figure 2) [6]. This highlights the need for expert guidance regarding treating carbapenem-resistant infections. Interestingly, fluoroquinolones, especially ciprofloxacin, are the only class of oral antibiotics with specifically antipseudomonal activity. Regardless, for those with risk for serious infections, combination therapy with agents from different drug classes is generally suggested. Most commonly, a combination of a beta-lactam along with an aminoglycoside is chosen [3,7]. However, resistance to these standard antimicrobials is rapidly growing, particularly in hospital-acquired isolates [8]. In fact, 2.8 million cases of multidrug-resistant bacterial infections were reported in the U.S. alone in 2019 [9]. Unfortunately, although using two classes of drugs synergistically has proven to be helpful, it has not solved the problem of multidrug-resistant, extensively drug-resistant, or pandrug-resistant strains. Approaching pathogens with an “out of the box” approach, such as that explored in this review, may be necessary to aid the antimicrobial strategy of treatment in *P. aeruginosa* infections [3,10].

In the ongoing battle between humans and the pathogenic microbes that cause disease, the CDC recognizes that the development of newer antimicrobial pharmacotherapeutics continues to be a pressing need, despite several current pharmaceutical agents that are reserved for the treatment of multidrug-resistant isolates [7]. In response to advancing antimicrobial pharmacotherapies, particularly bactericidal therapies that impose selective pressure, bacterial resistance mechanisms continue to evolve as opportunistic microbes adapt to an ever-changing therapeutic landscape. The evolution of multidrug-resistant *P. aeruginosa* can be considered as a case study based on its sophisticated *quorum sensing* communication system and phenotypic plasticity that has allowed it to adapt, survive, and thrive in a wide variety of environmental (e.g., aquatic and soil) and host conditions [11]. Among clinical isolates, a wide range of phenotypic variation has been identified including hyperpigmentation, small colony variant formation, autoaggregation, alginate overproduction, and autolysis [12,13,14,15,16]. These phenotypes change and adapt as an infection progresses, allowing for long-term survival in the differing conditions of the host [17]. The exact number of clinically significant strains of *P. aeruginosa* is unknown, but up to 40 individual strains were identified in the late 1970s, and as of 2019, there have been at least 66 clinical strains and 19 environmental strains identified [18,19]. This versatility has led *P. aeruginosa,* which is prone to the development of antibiotic resistance that renders current treatment options inadequate [2,8], to be categorized as part of the ESKAPE (*Enterococcus faecium Staphylococcus aureus*, *Klebsiella pneumonia*, *Acinetobacter baumannii*, *P. aeruginosa*, and *Enterobacter* species) group of bacteria, deemed of particularly great health concern by the Infectious Diseases Society of America (IDSA) [20]. Microbial evolution to circumvent and resist antimicrobial therapies necessitates distinctly new approaches to targeting multi-resistant bacterial species, which cause human disease and place a large burden on the healthcare system.

Antibiotic resistance continues to increase through the development of beta-lactamase-producing strains, shifts in porin conformations, efflux pumps, specific antibiotic inactivating enzymes, and non-specific porin modifications [8,22]. A deeper understanding of the mechanisms of resistance can inform drug discovery and development efforts. A central mechanism in the development of antibiotic resistance is the conversion to coordinated communal biofilm growth. As commensal *P. aeruginosa* changes to an opportunistic pathogen or acquired pathogenic infection is established, genotypic and phenotypic changes occur. Among the most important, *P. aeruginosa* switches from a more antibiotic susceptible, motile planktonic phase to an antibiotic tolerant, non-motile biofilm phenotype. This switch represents a key element in the conversion to pathogenicity. Importantly, this mucoid, biofilm phenotype is seen in over 75% of CF-associated *P. aeruginosa* infections [23,24]. Once in its sessile phase, *P. aeruginosa* can attach to host epithelial cell layers to better avoid the host’s immune responses [2]. Although anti-*P. aeruginosa* IgA antibodies are also present lining the respiratory tract, limiting the spread of *P. aeruginosa* to the respiratory epithelium, they have shown to be ineffective at eliminating infection [25]. Interestingly, antibiotic use may cause IgA deficiency by inhibiting normal microbiota and Toll-like receptor (TLR) signals that induce activating factors of IgA production in plasma cells. Therefore, excessive antibiotic use may further increase the risk of *P. aeruginosa* infection or reinfection [25,26]. These problems underscore that passive immunity may provide the safest and most efficacious therapeutic and even potentially prophylactic option. A thorough analysis of the varied mechanisms of antibiotic resistance is beyond the scope of this review, but some of the types of drug resistance mechanisms in *P. aeruginosa* are shown in Figure 3 for reference.

Chronic bacterial infections are often heralded by the presence of a subpopulation of persister cells, bacterial cells capable of surviving an antibiotic assault. This subpopulation of bacteria is typically present in high numbers in bacterial biofilms [26]. Although initially their ability to persist in the presence of a bactericidal agent may be indicative of some innate antibiotic resistance, a deeper dive into the mechanism suggests that this population is able to decrease their metabolic and growth activity to a state of senescence, circumventing antibiotic action, which often requires active metabolism or proliferation. According to this hypothesis, as borne out by numerous studies, persistence may be a transient state, which may or may not be passed to subsequent generations based on the underlying mechanism of action. Alternatively, several genes have been identified (e.g., hip-high persistence [27] and GlpD—sn-glycerol-3-phosphate dehydrogenase [28]) and have been suggested to confer phenotypic persistence, a characteristic which may prove heritable [29,30,31]. The persister cell’s phenotypic switch, which is likely the result of complex signaling cascade(s), between metabolically quiescent and metabolically active may prove to be an incredibly important drug target; however, an in-depth discussion of the variability and complexity is beyond the scope of the current paper and the reader is referred to several good recent reviews of the topic, specifically that from Kaldalu et al., in 2020 [32] and Louwagie et al., in 2021 [30].

*P. aeruginosa* isolates taken from cystic fibrosis patients with chronic colonization have shown high levels of persister cells, one of the leading hypotheses behind the inability of antibiotics to effectively clear colonized bacteria [26,33,34]. While intrinsic antibiotic resistance mechanisms lead to multidrug-resistant acute *P. aeruginosa* infections, chronic infections may be more likely to display antibiotic-tolerant mechanisms such as these persister cells and biofilms [33,34]. Many of the quorum sensing mechanisms to be described later may underlie the development and proliferation of these persister cells, although these processes remain poorly understood [26,35].

## 2. Host Immune Response

In addition to using mechanisms of *P. aeruginosa* antibiotic resistance to inform and frame drug development efforts, knowledge of the host immune response to developing *Pseudomonas* infection can also inspire drug discovery and new clinical treatment strategies. A brief discussion of the host/microbe interaction and immunity provides necessary context.

*P. aeruginosa* infection commonly induces a robust humoral response including IgG antibodies towards lipopolysaccharide (LPS), alginate, alkaline protease, elastase, exotoxin A, and many other surface antigens and proteins of *P. aeruginosa*, which are often upregulated as virulence factors during various stages of biofilm development [36,37]. Unfortunately, the host antibodies produced typically have low affinity for their respective targets and are not effective at eliminating the infection [25]. As an aside, host opsonizing antibodies also cannot eliminate these mucoid microorganisms [38]. Nevertheless, anti-*P. aeruginosa* IgG binds to its antigen and immune complexes are formed, activating complement and recruiting macrophages. As macrophages and immune cells bind the anti-*P. aeruginosa* IgG, they create reactive oxygen species (ROS), consuming oxygen, making the biofilm environment more anaerobic and thus more favorable for the organism. The anaerobic environment is unsuitable for host macrophages, neutrophils, and other immune cells. Phagocytosis may occur, but without sufficient oxygen, ROS cannot be produced to eliminate bacteria. This creates an inflammatory environment causing tissue damage without efficient disruption of *P. aeruginosa* biofilms [25]. The ongoing inflammatory state in chronic infections is thus not linked to immunogenicity of the bacterial organisms themselves, but rather the secreted products that leave the biofilm and induce immune responses in the airway epithelium [23,24]; hence, the humoral responses produced by many people in response to *P. aeruginosa* infection are not effective in eliminating the infection.

The lack of a humoral immune response capable of eliminating *P. aeruginosa* has a significant impact on vaccinologists’ ability to actively provoke the immune system with a *Pseudomonas* vaccine. Vaccines against *P. aeruginosa* have been developed and tested with the same unsatisfactory effect. While successful at producing antibodies, to date, vaccines have been unsuccessful at preventing infection in many individuals [39]. For example, a recent clinical trial studied the efficacy and safety of IC43 vaccination against *P. aeruginosa* in ICU patients. IC43 contains the *P. aeruginosa* outer membrane proteins OprI and OprF. Findings suggest that while the vaccination is highly immunogenic, the candidate vaccine did not reduce overall mortality [40]. Further work here, specifically on target identification, is warranted.

## 3. Description of Targets

Taking inspiration from the immune response and in the context of *P. aeruginosa*’s life cycle and its antibiotic resistance mechanisms, several potential targets secreted by *P. aeruginosa* were identified. These targets, outlined in Table 1, produce a wide variety of effects in hosts and the bacteria, contributing to the pathogenesis of the entire spectrum of infections caused by this organism. Some of the most promising targets identified are discussed in the following sections; however, a more complete list can be found in Table 1. Identified targets are grouped into eight categories based on their function and location in the cell.

### 3.1. Secreted Toxins and Invasins

Targeting extracellular-secreted toxins and invasins may have the greatest potential to decrease virulence and improve clinical outcomes. “Secreted toxins,” here, is not referencing toxins injected into the cytoplasm of host cells (i.e., intracellular toxins) which are not highly available to immunotherapeutics. Targeting secreted extracellular toxins can be impactful not only to the host, but also to commensal flora. Invasins penetrate host cells, allowing entry during the initial stage of infection. Commonly, these extracellular toxins and invasins are available to interact with circulating and secreted (e.g., IgA and IgM) antibodies.

Hence, this class has several very promising targets for antibody therapy development. Additionally, they do not impose “life-or-death” selective pressure, allowing therapeutic efficacy to be preserved beyond what is common for many antibiotics, which experience emergence of resistance within a short time of their entrance to the market [31,37]. Several common members of the toxin/invasin class, which are (1) ubiquitously and extracellularly expressed or secreted, (2) key mediators of virulence and pathogenicity, (3) responsive to antibodies, and (4) specific to *P. aeruginosa*, are discussed below and should be strongly considered as a part of any therapeutic antibody strategy.

**Exotoxin A.** Exotoxin A is the most potent virulence factor produced by most clinical strains (88%) of *P. aeruginosa* [41]. Exotoxin A is a ribosylating enzyme that inhibits protein synthesis and interferes with immune functions of the host and causes widespread apoptosis [42]. The regulation of exotoxin A is not completely understood, although it is thought to be upregulated for iron scavenging.43 However, it is known that exotoxin A is secreted through the Type II Secretion System (T2SS, see Figure 3), making it a promising target for therapeutic antibody development [43]. In several studies, exotoxin A antibodies have provided protection to clinical *P. aeruginosa* infections, including in exotoxin A toxoid vaccine trials [44].

**Protease IV.** Proteases are associated with the corneal damage seen with *P. aeruginosa* keratitis and play an important role in the tissue damage seen with soft tissue *P. aeruginosa* infections [45]. Protease IV, which acts to cleave or degrade host proteins such as immunoglobulins, complement, and fibrinogen, is a somewhat unique extracellular virulence factor [46]; its expression and potency are induced by *quorum sensing* (see below) [47]. While host-derived antibodies to Protease IV fail to develop in acute infections, injection of antibody–antigen complexes has been shown to develop strong, protective antibody responses [48,49]. Alternatively, antibody inhibitors of Protease IV could be developed as medications for *P. aeruginosa* infections. In one study, an antibody inhibitor showed complete inhibition of the protease [46].

**Lipase A (LipA).** Lipase A is abundant and a major secreted protein of *P. aeruginosa* [50]. In fact, Lipase A is the main extracellular lipase of *P. aeruginosa*. LipA is transported across the cell envelope by a type II secretion system, which is a two-step ATP-dependent process. It has been shown that Lipase A binds to the extracellular polysaccharide, alginate, by electrostatic interaction [50]. This interaction seems to localize the enzyme near the cell surface and is thermo-stabilizing, which may be relevant for the growth and survival of biofilm resident *P. aeruginosa* [50]. LipA is also an immunomodulator [51,52]. Therefore, therapies that include Lipase A as a target would help decrease the establishment of infection and decrease virulence.

**Phospholipase C.** Phospholipase C (PLC), also known as lecithinase, is a common class of enzymes that cleave exogenous phosphatidylcholine (PC) into fatty acids and choline. *P. aeruginosa* uses the processed phospholipid products as an energy source. *P. aeruginosa* has two types of PLC, hemolytic (PLCh) and nonhemolytic (PLCnh). The hemolytic one seems to play a larger role in virulence [42]. It is hypothesized that PLCh decreases oxidative burst activity in neutrophils, which are the primary cells responsible for clearing *P. aeruginosa* from the lungs. In mouse models, PLCh has also been shown to cause an increase in vascular permeability, end-organ damage, and death [53]. *P. aeruginosa* strains with either disruptions or deletions in PLCh are less virulent than the wild-type strain [53]. While commonly a membrane-bound protein, in *P. aeruginosa* infections, PLC is a secreted toxin that plays a role in pathogenesis [54]. PLC’s ability to damage the host cell membrane allows disseminated infection and wound colonization, indicating that PLC may play an integral role in the establishment and maintenance of chronic wound infections [41]. Virulence factors expressed by *P. aeruginosa* isolates from chronic leg wounds include beta hemolysins (92.3%), lipase (76.9%), and lecithinase (61.5%) [41]. Additionally, a study of 123 environmental and clinical strains of *P. aeruginosa* also expressed beta hemolysins (95.1%), lipase (100%), and lecithinase (100%), displaying high conservation of this class of enzymes [55]. This high degree of conservation and known genetics (i.e., ExoT and AlgD, coded for phospholipases and protease IV, respectively [41]), makes this a particularly attractive target for therapeutic enzymes [53]. Antibodies to PLC have been detected early and at high levels in many patients, but especially in CF patients with chronic *P. aeruginosa* infections. The antibodies remain elevated throughout the course of infection and increase with acute exacerbations [56]. The one particularly important caveat to targeting PLC for therapeutic antibody development is that while PLCh is secreted, PLC may be sequestered in secreted outer membrane vesicles, providing only limited availability [57]. Nevertheless, the potential of this class warrants further study.

**LasA and LasB (Elastase A and B).** LasA and LasB are two secreted proteases that cleave immunoglobulins, inhibit cytokines, interfere with immune cell functions, and increase the permeability of the tight junctions of the airway epithelium [58,59,60,61,62,63,64,65]. Both are synthesized as proenzymes. LasA is secreted in its unprocessed form and then processed extracellularly. LasA is known to enhance pseudomonal colonization and immune evasion via enhanced syndecan-1, IL-6 receptor, CD14, and TNF-α shedding, and facilitates the function of elastase [65]. LasA can also act on elastin but is limited to a few highly specific amino acid sequences [66]. In contrast, LasB is a zinc metalloprotease that can cleave proteins at multiple sites. LasB is a propeptide that is initially secreted and then partially degraded extracellularly. LasB is able to degrade a wide range of host proteins, conferring much more virulence when compared to LasA [45,66]. Part of this enhanced virulence is also the ability of LasB to induce damage to host tissue and subvert immune responses. In addition, LasB decreases the expression and activity of the CFTR ion channel in bronchial epithelial cells and has been shown to have the ability to degrade IL-6 and trappin-2, which are both important for antimicrobial defense [45]. LasB seems to exacerbate this weakened system and cause increased morbidity and mortality for cystic fibrosis patients [45]. LasB is prevalent and a major secreted protein in the secretome of the *P. aeruginosa* PA01 strain, driving virulence in a large percent of CF patients [45]. LasB is secreted by the type II secretion system (Figure 3) [45].

In addition to the central role of LasA and LasB in virulence, past studies of active immunity indicate that antibody-based therapeutics might be particularly effective. Anti-LasA and anti-elastase antibodies may decrease virulence by rendering *P. aeruginosa* more vulnerable to host immune mechanisms or antibiotics. Vaccine candidates to LasB and LasA were efficacious in preventing and decreasing the severity of pneumonia in minks, corneal ulcers and lung infections in rats and burns in mice [67,68,69,70]. LasA and LasB may be secreted in outer membrane vesicles and thus would have limited availability as drug targets [57].

**Alkaline Protease (AprA)**. AprA, which is also a zinc metalloprotease [71] that has activity and function similar to LasB, is a prevalent secreted protein of the *P. aeruginosa* PA01 strain [71]. Its production is thought to be regulated not only by phosphatidylcholine, but it appears that *P. aeruginosa* is also able to induce the production of alkaline protease in conditions of limited inorganic phosphate [72], tagging it as potentially relevant target to squelch virulence [73]. AprA is found to be active during an inflammatory phase of infection and causes cellular necrosis and hemorrhage via increased vascular permeability and proteolysis in infected wound sites. This activity is further expressed in septicemia, in which proteolysis affects homeostatic mechanisms of plasma proteins [74]. Unlike LasB, AprA is secreted as a part of the Type 1 secretion system which includes AprA, D, E, and F. Unfortunately, there is some evidence to suggest that Alkaline Protease may be sequestered in OMVs and thus may have limited availability [57]. Thus, AprX, a caseinolytic extracellular protease, also secreted by T1SS, might serve as a good alternative or supplementary target [75].

**Caseinase.** Caseinase is a major, soluble protease and virulence factor secreted by *P. aeruginosa* that breaks down casein [41], although there is some evidence to suggest that caseinase may be sequestered in OMVs after secretion, which may limit its utility as an antibody drug target [57]. In isolated strains of *P. aeruginosa*, caseinase was expressed in 91.67% of chronic leg ulcers cultured [41]. Another study analyzed 123 clinical and environmental *P. aeruginosa* strains and reported a 99.2% conservation of caseinase. Interestingly, the clinical strains had considerably more proteolytic activity compared to environmental strains [55]. In wounds, caseinase delays healing and contributes to chronic lesions by limiting essential amino acids in the area [41]. Additionally, caseinase and other secreted proteases aid in colonization, tissue damage, and immune evasion by *P. aeruginosa* [55]. Thus, therapies that include caseinase as a target would theoretically decrease the establishment of infection as well as the virulence of the strain.

### 3.2. Secretion System Proteins

While the previous section focused on virulence factors and other key molecules secreted, targeting key components of the secretion systems themselves may have the same functional consequence. These secretion systems traverse bacterial membranes and are classified into eight distinct types based on their specific structure, composition, and activity. Types I, II, III, V, and VI are particularly relevant due to their significance in the virulence of *P. aeruginosa* infections, although evidence suggests that *P. aeruginosa* possesses all eight types [76].

**Type 1 (T1SS) Secretion System.** Two well-characterized, independent type I secretion systems (T1SS) have been identified in *P. aeruginosa*. T1SS are associated with virulence and found to be active during the inflammatory phase of *P. aeruginosa* infection, 77 making them potential targets for pharmacotherapy. The Apr T1SS is an ABC transporter involved in alkaline protease secretion (AprA) and (AprX) [76,77]. The HasF T1SS is a heme-binding protein secretion system (HasF) [68,69]. AprF and HasF are the outer membrane proteins of the HasF T1SS involved in alkaline protease secretion and heme uptake [78,79] (see Figure 4), and as such would be accessible to antibody and other traditional therapeutics. The main question is the availability of these membrane proteins since their immunogenicity has yet to be characterized in detail [76]. Further immunogenicity and accessibility characterization is warranted. Two other T1SS have been identified using genomic analysis but are not well-characterized currently.

**Type II (T2SS) Secretion System.** The T2SS secretes major toxins (e.g., noted above in section A, including LasA and LasB, type IV protease, and exotoxin A) into the extracellular space during acute infections. Molecules secreted through T2SS secretions cause a significant number of the disease pathologies associated with *P. aeruginosa* infections [76]. XcpQ and HxcQ are outer membrane proteins of the two major T2SSs (see Figure 4), making them accessible to circulating (e.g., IgG and IgA) and secreted (IgA and IgM) therapeutic antibodies. Unfortunately, as noted with the T1SS membrane proteins, these membrane proteins have only limited characterization of their immunogenicity [80].

**Type III (T3SS) Secretion System.** The T3SS is the other major secretion mechanism for toxins. More accurately, the T3SS injects molecules, commonly DNA and toxins, into the cytoplasm of the host cells [81]. These T3SS toxins play a role in some localized infections, including the eye and lung [82,83,84]. While the toxins are not secreted extracellularly in many types of *P. aeruginosa* infections, antibodies to these toxins might be considered for some indications. The hallmark of T3SS is the needle complex (NC) or injectisome (described below and in Figure 4). The T3SS is most often identified with *Yersinia* [76,81]. However, the *P. aeruginosa* T3SS injects numerous exotoxins including ExoU,S,T,Y, MMP-12, and MMP-13 [85]. The injected toxins from T3SS alter the host’s cell cycle, commonly inducing apoptosis. The T3SS consists of a multimeric protein complex that is divided into four major domains. Epitopes in each of the extracellular domains could serve as drug targets to reduce bacterial associated morbidity. Other attempts to actively target the T3SS with antibody therapies have yielded mixed results [86], with enough promise that antibody therapy against T3SS, which in conjunction with T2SS is thought to significantly contribute to bacterial virulence [76], should be pursued.

**PcrV:PcrG.** PcrV and PcrG are heterodimers that are an integral part of binding the needle complex (NC)/T3SS apparatus (T3SA) of the T3SS to host cells. They are commonly found in the main secreted protein as part of the secretome of the *P. aeruginosa* PA01 strain and are anticipated to be available to circulating and secreted (IgA and IgM) antibody therapeutics. While PcrV’s role is not completely characterized, it (PcrV) seems to form a scaffold at the tip of the T3SS injection needle [76,87,88]. PcrV and PcrG are found in 100% of *P. aeruginosa* strains [89], while PcrV is specifically found in most *P. aeruginosa* strains associated with poor clinical outcomes [90]. In addition, the PcrV sequence and structure are associated with antibiotic resistance [90]. Vaccination to produce PcrV antibodies has been shown to be protective in mice infected with a lethal dose of virulent *P. aeruginosa.* Passive immunization with anti-PcrV IgG has also been demonstrated to be protective in animal models [91,92,93,94,95,96,97,98,99]. PcrV antibodies are accessible to the humoral immune system, as shown in clinical trials [92]. PcrG requires further study.

**PcsF.** PcsF is the needle part of the needle complex (see Figure 4). The molecule remains poorly characterized, although the PcsF is highly conserved even between *P. aeruginosa* and *Yersinia* [100]. Since PcsF is located extracellularly, it should represent an available target that should be pursued [100].

**PoP B&D.** PoPB and PoPD are part of the T3SS that forms a heterodimer on lipid membranes, allowing for penetration of target cell membranes by assembling functional translocons [101]. Their membrane location also makes them accessible to circulating and secreted (IgA and IgM) antibodies [101]. PoP is active during an inflammatory phase of infection. PopB and PopD are found in most *P. aeruginosa* strains associated with poor clinical outcomes [102].

**Type V (T5SS) secretion system.** T5SS predominantly secretes virulence factors and enzymes that support biofilm formation [103]. While the Type V secretion system (T5SS) involves a two-step process: synthesis of a precursor molecule and a periplasmic intermediate of the effector, it is considered the simplest secretion system [104]^.^

Since they are autotransporters, the epitope of the antibody would need to target the cleaved domain(s) of the transport protein. While EstA and LepA are potential targets associated with T5SS, PlpD is currently more characterized and thus has more potential as a drug target.

**PlpD.** PlpD is a phospholipase A1 enzyme in the patatin-like family that degrades lipids, especially in membranes [104]. PlpD is composed of multiple domains including a secreted domain that is fused to a transporter domain. Due to its high virulence and availability, it makes a solid therapeutic target.

**Type VI (T6SS) Secretion System.** T6SSs are effector translocation systems that resemble inverted bacteriophage-puncturing devices. Effectors of *P. aeruginosa* T6SS include Tse1-3, PldA, and PldB, well-known virulence factors [105]. Phospholipase D (PLD) enzymes, including PldA and PldB, are enzymes that catalyze the hydrolysis of phosphodiester bonds. In addition, T6SS machinery improves survival of *P. aeruginosa* by allowing better delivery of toxins to neighboring organisms; suppressing nonpathogenic normal flora may also help *P. aeruginosa* and translocate effector proteins directly into target host cells [106]. Hence, T6SS, which also plays a role in biofilm formation, can be considered a virulence factor of *P. aeruginosa*. T6SS consists of several protein components, namely hexameric rings of hemolysin-coregulated protein (Hcp), Val-Gly repeat (Vgr) proteins, and a PAAR protein. Hcp hexameric protein rings stack in vitro to form nanotubes that resemble bacteriophage tail tubes while Vgr proteins are similar to the tail-spike puncturing device of a phage. PAAR proteins are thought to aid in facilitating the puncture of target membranes [107]. Importantly, PAAR proteins as well as Hcp and Vgr complexes may constitute a portion of the surface-exposed T6SS machine [106,107], making them potential drug targets to reduce morbidity secondary to *P. aeruginosa* infection.

### 3.3. Quorum Sensing/Metabolites

In a process called *quorum sensing*, *P. aeruginosa* secretes small molecules to communicate local population density. As a result of these small *quorum sensing* (QS) molecules bacteria will modulate gene expression to change their mode of growth (i.e., planktonic vs. sessile), increase their virulence, improve their resilience in the face of antibiotics and other therapeutics, and act in a coordinated way to “benefit the group” [108]. Since QS plays a central role in virulence, the receptors and small molecules that coordinate these efforts have earned a spot on the shortlist for drug targets. This section offers insight into potential anti-virulence strategies using antibodies to small QS molecules and enzymes that produce them. Antibody development to small molecules is technically difficult; however, these molecules represent potentially highly efficacious therapeutic targets.

***Pseudomonas* Quinolone Signal (PQS; 2-heptyl-3-hydroxy-4-quinolone).** PQS is a ubiquitously expressed, small, intercellular signaling molecule that is actively expressed during infection. Its intercellular nature means that it would be accessible to circulating and secreted (IgA and IgM) antibody therapies. PQS regulates numerous virulence-related factors and functions [109] in *P. aeruginosa* and may participate in the biogenesis of OMVs [110]. The impact of PQS in surrounding tissues includes autolysis of damaged cells and decreased cellular metabolic activity. There is also evidence that PQS may act as a protective stress response signal, iron scavenger, and host immune modulator [110]. Finally, and perhaps most critically, PQS can both induce oxidative stress as well as provide an antioxidative response. It is these PQS functions that may prove most protective for *P. aeruginosa* against damaging oxidative bursts from polymorphonuclear cells. These mechanisms kill host tissue and provide selective pressure on bacteria in both chronic biofilm infections and in acute infections [109]. The critical protective nature of PQS means that although developing an antibody to PQS may be possible, it may lead to selective pressure and push *Pseudomonas* to evolve [109]. However, delivered properly, any therapeutic strategy that includes PQS could have a large impact on antibiotic-resistant *P. aeruginosa*.

**Pyocyanin.** Pyocyanin, or phenazine, which gives cultures their characteristic blue-green coloration, is a secreted virulence factor found at the site of infection in the host. Pyocyanin is a small-molecule toxin that is easily able to cross cell membranes usually through a T2SS quorum-sensing dependent mechanism. Pyocyanin is a redox-active tricyclic zwitterion shown to affect the respiratory, cardiovascular, urological, and central nervous systems by inducing oxidative stress within cells [111,112]. Pyocyanin directly oxidizes and depletes reduced glutathione, effectively neutralizing one of the major redox pathways. Additionally, pyocyanin induces IL-8 and LTB4, which attracts neutrophils, and although it seems counterintuitive, the compound then inhibits IL-2 and IL-2R expression, causing neutrophil and lymphocyte apoptosis. Neutrophil apoptosis releases proteases, leading to host tissue destruction and inflammation. Due to this cascade, pyocyanin released by *P. aeruginosa* is especially toxic to cystic fibrosis patients, potentially due to its ability to induce neutrophil extracellular traps (NETs), which are able to exacerbate airway inflammation for their own benefit [113]. Thus, pyocyanin is a powerful mediator of cell injury used by *P. aeruginosa* and is a prime target to halt disease [112]. In fact, in vitro studies have demonstrated that anti-pyocyanin antibodies can provide protection [114]. However, just as with PQS, although antibody development to small molecules is often feasible, it is difficult. Since pyocyanin is secreted, it is available for binding by both circulating and secreted (IgA and IgM) antibodies. As with endotoxin A, pyocyanin is ubiquitous, available, and pathologic, and should be explored as a target.

**N-(3-oxododecanoyl)-l-HSL(3-oxo-C12-HSL/OdDHL).** 3-oxo-C12-HSL, a specific type of acyl homoserine lactone (AHL), is an auto-inducing, quorum-sensing associated virulence factor that regulates swarming, toxin and protease production, and proper biofilm formation [115,116]. OdDHL inhibits naive T-Cell proliferation as well as subtype differentiation. It acts as a *quorum sensor* signal molecule and inhibits IL-12 and TNF [117]. It also decreases antibody production at high concentrations and promotes IgE/IL-4 at low concentrations. OdDHL also regulates Las gene expression, most notably LasR, and upregulates IL-8 in corneal infections [115,116]. Additionally, it has been shown to inhibit dendritic cell concentrations in a dose-dependent fashion. 3-oxo-C12-HSL inhibits PPAR gamma functioning in host cells, leading to an active, expanding proinflammatory state and bacterial swarming. Low-dose antibiotics can suppress the *quorum sensing* functions of 3-oxo-C12-HSL; however, 3-oxo-C12-HSL can auto-induce Las systems, producing cytotoxic effects targeting macrophages and neutrophils via pro-apoptotic pathways. This molecule also selectively regulates Nf-kB signaling, leading to decreased host TLR4 pathway utilization to fight *P. aeruginosa* infections [118,119,120,121].

Evidence suggests that mice immunized with a carrier-conjugated 3-oxo-C12-HSL were able to generate a protective humoral response. Thus, as with other QS molecules, antibody generation, while challenging, is possible. Since 3-oxo-C12-HSL is secreted, it is available for targeting and binding by circulating and secreted (IgA and IgM) antibodies [122].

**Hydrogen cyanide (HCN)**. HCN production is an important mediator involved in *quorum sensing* (QS). HCN-producing bacteria help to maintain cooperativity and eliminate “cheating bacteria” [108]. Bacteria that participate in QS have increased resistance to HCN intoxication and ROS damage than the mutant cheaters [123]. These cheaters threaten cooperativity and thus stability of the *P. aeruginosa* bacterial population. HCN has been detected in sputum cultures of *P. aeruginosa*-infected CF patients and *P.aeruginosa*-infected burn cultures. Researchers postulate that HCN decreases pulmonary function, especially in CF patients, by interfering with essential enzymes, such as superoxide dismutase, NO synthase, cytochrome C oxidase, and others, disrupting aerobic respiration and cellular immune functions [124]. In burn patients, HCN inhibits cytochrome c oxidase to decrease metabolism at the site of infection [125]. Targeting HCN production may disrupt the delicate nature of *P. aeruginosa* biofilms, helping to treat resistant infections [108,123].

To decrease HCN signaling, two targets are possible: (1) HCN, which is difficult given the tiny size of the HCN molecule or (2) HCN synthase, a membrane-bound enzyme that synthesizes HCN from glycine [125]. Targeting HCN synthase, which is a ubiquitous, membrane-bound molecule, available to circulating and secreted (IgA and IgM) therapeutic antibody delivery, may prove to be a viable strategy for combating QS.

**Pyomelanin.** Pyomelanin is a small molecule in the same family as melanin that is secreted as a part of the *P. aeruginosa* QS system. Pyomelanin overproduction is a common phenotype among patients with cystic fibrosis and urinary tract infections, giving colonies a brown phenotype and heightened resistance to phagocytosis. Production of pyomelanin is associated with an increased resistance to peroxide [126].

Since pyomelanin is a small molecule that is secreted, it is antibody accessible. However, synthesis of antibodies to pyomelanin may not be a viable strategy due to the difficulty in production; nevertheless, if an antibody system could be developed against pyomelanin, it may be effective based on the accessibility of the antigen [127].

**Other candidate molecules.** As more information is gathered, several other quorum-sensing molecules may emerge as targets [128], including the Acyl Homoserine Lactones (AHLs): OdDHL, ConA, and BHL which are produced by the majority of *P. aeruginosa* strains. These molecules increase the organism’s virulence and have a wide array of immunomodulatory effects. Generally, AHLs are modulated and expressed secondary to a variety of *quorum sensing* signals. The variety of AHLs work in conjunction to divert the immune response away from the *P. aeruginosa* organism [123,129]. AHLs suppress T-cell growth and proliferation, particularly in CD4+ T-cells. They also promote Th2 over Th1 host immune responses. These molecules could serve as antibody-based drug targets.

### 3.4. Antibiotic Resistance Determinants

Antibiotics remain the best tool to fight *P. aeruginosa* infections. Unfortunately, as noted above, antibiotic resistance reduces antibiotic effectiveness. Targeting antibiotic resistance mechanisms with antibody-based therapeutics may resurrect some antibiotics, create synergistic new combination therapies, and expand the toolbox available to clinicians in the arms race against opportunistic pathogens.

**SecYEG and SecA.** Antibiotic resistance may be in part due to the Sec proteins’ functions as efflux pumps [76]. Hence, targeting the Sec proteins may be an effective countermeasure in the fight against antibiotic resistance. Upregulated efflux pumps, which can effectively eliminate intracellular acting antibiotics or other cytotoxic compounds by pumping them out of the cell and into the periplasmic space, are a main driver of antibiotic resistance [130].

The Sec proteins are all outer membrane accessible and thus potentially available as drug targets. In addition, these proteins are conserved in most antibiotic-resistant *P. aeruginosa* strains [76]. Sec are potentially accessible to circulating and secreted (IgA and IgM) therapeutic antibodies. Since Sec proteins are ubiquitous and pathologic, any therapeutic strategy could include these targets [76]. As noted with the T1SS membrane proteins, these membrane proteins have only limited immunogenicity characterization.

**OprM**. OprM is a protein on the tip of an efflux pump that specifically confers *P. aeruginosa* antibiotic resistance. The MexAB-OprM efflux system is in the resistance-nodulation cell division (RND) family of exporters [131,132], which includes 12 total RND pumps identified in the *P. aeruginosa* PAO1 genome [133]. MexA and MexB proteins complex with the outer membrane porin-like OprM to form a one-step efflux pump responsible for resistance to a large number of antibiotic classes, such as β-lactams, β-lactamase inhibitors, fluoroquinolones, tetracyclines, novobiocin, thiolactomycin, sulfonamides, macrolides, aminoglycosides, etc. [134].

As a ubiquitous outer membrane protein, OprM protein is accessible and thus potentially available as a drug target [135], although the exposed part is a relatively small part of the overall protein and prone to escape mutations. Nevertheless, OprM proteins are conserved in most *P. aeruginosa* strains that are antibiotic resistant. As noted with other membrane proteins, OprM has only limited immunogenic characteristics.

### 3.5. Other Membrane Biomolecules

Beyond conferring antibiotic resistance, *P. aeruginosa* has many other cell surface molecules with a variety of functions. These functions include regulation of swarming motility, host cell modification, biofilm composition, and adhesion, all of which are promising potential targets for antibody therapies. Unfortunately, generally speaking, these biomolecules are difficult to target as they have evolved to evade immune systems and are commonly poorly immunogenic; however, engineering antibodies to these targets could prove a fruitful endeavor if successful. The following section explores some of the most promising candidates.

**LecA and LecB.** The ability of *P. aeruginosa* to adhere to host cells is essential to pathogenesis. Adhesion is mediated by pseudomonal lectin receptors to glycoproteins on the surface of the host cell. LecA and LecB are located in the cytoplasm of *P. aeruginosa* and bind to galactose and fructose. LecA has been shown to have a cytotoxic effect on respiratory epithelial cells by decreasing their growth rate. It has also been shown to alter the permeability of intestinal epithelium to allow for increased absorption of other secreted toxins such as exotoxin A. LecB has been shown to play a role in pilus biogenesis and protease IV activity increasing virulence [136].

In mouse studies, administration of carbohydrate inhibitors such as methyl-d-galactoside (for LecA) led to the highest rates of survival, while administration of α-methyl-l-fucoside (for LecB) did not seem to impact survival. The lectin inhibiting carbohydrates also reduced the rates of lung injury and showed an increase in bacterial clearance from the lungs in the study population [136]. These results highlight the potential of LecA as a drug target; however, LecB requires further study to assess its potential value as a drug target.

**Phosphatidylcholine (PC) Metabolism and Transporting Enzymes.** Phosphatidylcholine (PC), a highly prevalent lipid found in lung surfactant, has been identified as a nutrient source for *P. aeruginosa* [54]. To serve as a nutrient source, PC is cleaved by *P. aeruginosa* PLC into fatty acids, glycerol-3-phosphate and glycerol. Numerous enzymes then transport these metabolic products into the bacterial cell. Glycerol-3-phosphate transporter (GlpT) and glycerol uptake facilitator protein (GlpF) are transmembrane proteins involved in the uptake of glycerol-3-phosphate and glycerol, respectively [137]. Blocking the metabolism of PC (as previously discussed for PLC) and/or transport of these degraded components of PC should result in decreased replicative potential and biofilm development of *P. aeruginosa*. Therefore, these metabolites and their transporters pose great targets for pAbs [54]. Notably, there are other key transporters that may also provide effective targets. Two choline symporters, BetT1 and BetT2, as well as an ABC transporter, CbcXWV, are involved in choline transport into the bacterium [138], Fatty acid transporters, such as long-chain fatty acid translocase (FadL), are utilized for free fatty acid uptake by *P. aeruginosa* [54,137]. Blocking any of these transporters may have a similar effect as blocking GlpT and GlpF.

**Multivalent Adhesion Molecule 7.** MAM7 is a protein that binds to the host cells during the early stages of infection. MAM7 binds with fibronectin and phosphatidic acid of the host’s cell membrane. Hence, MAM7 expression in *P. aeruginosa* is necessary for virulence. In a recent study of burn patients, topical application of polymeric microbeads functionalized (covalently attached) with the adhesin MAM7 led to positive clinical outcomes, including reduction in *P. aeruginosa* bacterial loads by decreasing the available receptors for the bacteria to attach in the wound and thus preventing bacterial spread [139]. The MAM7 protein is outer membrane accessible and conserved in most *P. aeruginosa* strains [139]; therefore, it may prove an effective therapeutic target.

**OprE and OprF.** OprE and OprF are major outer membrane proteins in *P. aeruginosa* that are associated with virulence, including cellular adhesion, virulence protein secretions via T3SS (e.g., ExoT and ExoS), and secretion of the quorum sensing-dependent virulence factors pyocyanin, elastase, lectin PA-1L, and exotoxin A. OprE and OmpA, are important in biofilm development. A mutation in the *ompA* gene resulted in a thinner exopolysaccharide layer of the biofilm [140]. Other surface membrane proteins are also involved in biofilm development. OprE and, additionally, OmpH, interact with OmpA to aid in biofilm development. OprF also binds to IFNγ, thereby increasing virulence [140]. OprE and OprF are also found in secreted outer membrane vesicles (OMVs) [141] and may be good therapeutic targets based on their accessibility [142]. However, as noted with the secretion system membrane proteins, the availability of these membrane proteins has only limited immunogenic characteristics, which may limit the development of therapeutic antibody-based treatments using traditional antibody technology. Nevertheless, the variety of proteins involved in biofilm development allows for the development of a multifaceted approach to *P.aeruginosa* antibiotic resistance These different biofilm components contribute to it being a suitable drug target for future pharmaceutical studies [143]. Further study is warranted.

**Lipopolysaccharide** Lipopolysaccharide (LPS) is a major component of the outer membrane of Gram-negative bacteria. For *P. aeruginosa,* LPS is a key virulence factor and stimulates an inflammatory response in the host. It contains three primary structural domains: lipid-A, variable O-antigen, and the core oligosaccharide, which can further be divided into the inner and outer core. The core oligosaccharides link the lipid-A domain, which is embedded into the outer membrane with the O-antigen, which in turn extends from the cell to interact with the host environment. Lipid-A can have a variable structure with differing levels of virulence and binds to the TLR-4 receptor, which is essential for LPS recognition via lipid-A, to stimulate the host’s innate immune system. The inflammatory response of the host increases the synthesis of LPS binding protein (LBP). The response that follows the binding of TLR-4 is dependent on the level of acetylation of lipid-A, which can be penta- or hexa-acetylated. In general, the hexa-acetylated form of lipid-A results in a much more robust inflammatory response.TLR-4 can also mediate host resistance to infection [144,145].

There are two types of LPS, smooth and rough. The presence of the O-chain determines whether the strain is considered rough. Rough LPS strains that are unable to produce the O-antigen are present in the lungs of CF patients. The smooth LPS isolates are responsible for systemic *P. aeruginosa* infections [144]. There have been attempts to develop a vaccine against LPS for *P. aeruginosa*. A conjugate vaccine was trialed on CF patients and showed initial promise. However, after moving on to phase III trials, the vaccine did not produce a strong enough immunological response [146]. Newer vaccines being trialed are attempting to elicit a stronger response. With incidence of *P. aeruginosa* antibiotic resistance becoming more prevalent, an effective vaccine would be a game changer in the fight against the bacteria [144,145]. Again, generating an antibody (or antibodies) to LPS is a difficult task as LPS is both a non-protein target and has variable regions; however, the payoff for such a development would be substantial as LPS is found in all *P. aeruginosa* strains and potentially virtually all Gram-negative bacteria.

**Rhamnolipids (A, B, C, G, R).** In addition to LPS, which is a hallmark of Gram-negative bacteria, *P. aeruginosa* also produces glycolipids called rhamnolipids with a rhamnose head group (glycosyl) and fatty acid tails. These membrane-bound glycolipids provide surfactant properties. The production of rhamnolipids is variable based on the *quorum sensing* that regulates Rhl gene expression. Rhamnolipids are found on the cell surface, in biofilms, and excreted into the environment. In fact, rhamnolipids are also found in the sputa of people with cystic fibrosis with *P. aeruginosa* infection. The molecules have a role in causing respiratory cell lysis and ciliary dysfunction in these patients by removing ATP-ase dynein arms from cilia [147]. While the function of rhamnolipids is speculative, it includes the uptake of hydrophobic substrates, improved motility, and mediating biofilm growth. Rhamnolipids are particularly important in modulating the swarming motility of colonies, which may serve to perpetuate bacterial survival and virulence. Reducing *P. aeruginosa* cell tension and improving the organism’s adherence to host cells may be inextricably connected to the rhamnolipid’s ability to modulate the bacteria’s swarming behavior. As a multi-purpose molecule, targeting rhamnolipids may carry biological and therapeutic benefits, but may also incur a host of side effects [148]. Nevertheless, in a drug targeting context, the most relevant functional aspect of rhamnolipid may be its ability to disrupt epithelial cell tight junctions, as evidenced by the presence of rhamnolipids in both lung and corneal infections [149,150].

As noted above with LPS, the development of antibodies to rhamnolipids, as is the case with many non-protein antibodies, is challenging for numerous reasons; however, antibodies have been previously developed [151]. Rhamnose-binding lectins may also be an option. Since rhamnolipids are membrane-bound, they have at least some epitopes available for antibody or lectin therapy delivery. Since rhamnolipids are ubiquitous, available, and potentially pathologic, any therapeutic strategy should include this target.

**Psl EPS/Mucoid Exopolysaccharide (MEP).** Psl exopolysaccharide (EPS/MEP) is a major component of the biofilm produced by *P. aeruginosa*. It is produced by several enzymes with increasing expression in sugar-rich environments. Psl EPS is composed mainly of galactose and mannose and contributes to the matrix of the biofilm by initiating and strengthening biofilm adhesions. The biofilm decreases antibiotic penetration into *P. aeruginosa* allowing survival and propagation of antimicrobial-resistant strains [152]. As with rhamnolipids and LPS, EPS is a valuable target, but it is difficult to generate antibodies for numerous reasons. Nevertheless, antibodies have been previously developed. CF patients often generate anti-EPS antibodies; however, these antibodies are non-opsonizing and non-neutralizing [153]. Using a FAb (antigen-binding fragment of an antibody) and recombinant opsonizing Fc (tail region of an antibody that interacts with cell surface receptors) to engineer synthetic antibodies could improve efficacy in order to clear the infection [154]. Alternatively, as with other carbohydrate targets, lectins may also be an option. Since EPSs are an integral membrane component, they are available to circulating and secreted (IgA and IgM) antibody or lectin therapeutic delivery. Since EPS is ubiquitous, available, and potentially pathologic, any therapeutic strategy could include this target.

**Alginate.** Alginate is another exopolysaccharide produced by *P. aeruginosa.* It spans the organism’s inner and outer membrane. Alginate enhances cell to cell adhesions and cell to host adhesions. These adhesions are protective for the organism; they increase resistance to the host immune system [155]. *P. aeruginosa* evades the host immune system during chronic disease by decreasing its virulence so as to not provoke an acute immune reaction. Alginate is produced in large quantities during chronic infections. Inhibition of macrophage clearance and efferocytosis have been observed, but the mechanism of alginate in this process is still unknown [156]. Again, as with other lipids/carbohydrates discussed in this section, alginate is a difficult, but potentially valuable, target.

**Secreted outer membrane vesicles (OMVs).** OMVs are ubiquitously generated and secreted by *P. aeruginosa* during all stages of the bacteria’s life cycle [141,157], particularly under conditions of stress. Growing experimental evidence suggests that OMVs are secreted with numerous virulence factors, antibiotic resistance molecules, and *quorum sensing* molecules, including hemolysin, phospholipase C, elastase, and alkaline phosphatase, antimicrobial quinolines, adhesions, CIF, and beta-lactamase into the host cells [158,159,160,161]. Additionally, these vesicles carry virulence factors that can cause cytotoxicity, increase adherence, and carry factors such as cystic fibrosis transmembrane conductance regulator inhibitory factor [141]. OMVs can then fuse with host cell membranes and alter their function, resulting in the breakdown of the host’s epithelial barrier cells, which can ultimately facilitate the invasion of *P. aeruginosa*. Antibody therapies for these targets could include several approaches. First, antibodies to CIF could be delivered inside the vesicle that fuses to the secreted outer membrane vesicles. Second, antibodies to the OMV itself could be developed with an opsonization effector function to eliminate the OMV entirely. More study into OMVs as either a drug target or natural drug delivery system is warranted.

**CbpD.** CbpD is one of the conserved outer membrane vesicle proteins found in the majority of studied strains of *P. aeruginosa*. Due to its conservative expression and extracellular presence, CbpD may be a possible target of immunotherapy against the outer membrane vesicles and the virulence factors, which they carry [162].

### 3.6. Motility Factors

Bacterial motility plays a key role in *P. aeruginosa* virulence and biofilm formation. *P. aeruginosa* uses type IV pili and flagella for motility. Several studies have suggested that targeting pili or flagella with drugs or vaccines would show strong efficacy [163].

**Pili proteins. Type IV pili (TFP).** TFP is a motorized fimbriae providing twitching motility for *P. aeruginosa* [164,165]. TFP is essential to both biofilm formation and virulence. Several adhesin proteins (the minor pilins) are at the tip of the pili including PilEXWV and FimU (see Figure 3). The pilus is extended through major pilins and includes PilA [165]. Pili may also enable *P. aeruginosa* to perform transformation and conjugation, and exchange virulence and antibiotic resistance genetic material with other bacteria and may at least partially explain the organism’s ability to rapidly acquire antibiotic resistance.

The TFP proteins are all outer membrane accessible and are conserved in most *P. aeruginosa* strains [166]. TFP proteins are potentially accessible to circulating and secreted (IgA and IgM) antibody therapy delivery. Since TFP proteins are ubiquitous and pathologic, any therapeutic strategy could include these targets.

**Flagellar Proteins.***P. aeruginosa* acquires its motility from a single glycosylated polar flagellum. In addition to motility, the bacteria’s flagellum can play a role in adhesion to host cells and stimulation of the immune system. The flagellum is an emerging drug target due to the many roles it plays in the pathogenesis of *P. aeruginosa* [167]. There are two common types of flagella proteins, PAK and PA01. Antibodies against PAK flagella and PA01 flagella were used to assess their potential as a future drug target [168]. There have been varying results on the effectiveness of a vaccine targeting the *P. aeruginosa* flagellum. Rabbits were passively immunized with antibodies against the monomeric flagella protein and survival studies showed the antibodies provided little protection. However, antibodies targeting polymeric flagella showed promising results with an 87% survival rate following the *P. aeruginosa* challenge. Although anti-flagella antibodies against the PAK and PA01 strains showed only modest protection, flagella proteins appear to be a promising emerging drug target [168].

### 3.7. Resource Scavenging Molecules

One of the key battlefronts between host and bacteria is molecular resource scavenging. Iron is a prime example as it is an essential nutrient for the biological processes of *P. aeruginosa*. Like host cells and other bacteria, *P. aeruginosa* requires iron as a cofactor for many enzymes. It uses specialized iron scavenging molecules, siderophores, to steal iron from its environment. Antibodies to these molecules commonly provide protection against virulence and thus these molecules are excellent drug targets [169].

**Pyoverdine.** Pyoverdine is the major siderophore produced by *P. aeruginosa* that extracts iron from host complexes [170]. Pyoverdine has such a high affinity for iron that it can steal iron from transferrin and extract iron from host cells, leaving them metabolically limited and damaging mitochondria [171]. Pyoverdine is an extracellular peptide found in most *P. aeruginosa* infections, especially CF, and in most strains of *P. aeruginosa* [171]. As a conserved, extracellular protein [172], its accessibility to antibody development makes it an attractive candidate for therapeutic development.

**Pyochelin (Pch).** Pch is the second major siderophore secreted by *P. aeruginosa* to scavenge iron. Pch chelates iron and other metals from the ECM (extracellular matrix). Pch also provides a competitive advantage by inhibiting other bacteria, including *S. aureus* [173]. Pch is found extracellularly in most of the indications for *P. aeruginosa* infections and in most strains of *P. aeruginosa* [174,175]. As with pyoverdine, Pch is an extracellular protein and is conserved in most *P. aeruginosa* strains [175], making it a target worthy of consideration in the development of new therapeutic antibody strategies. Notably, Pch is a small molecule synthesized from cysteine and thus will require additional engineering to generate an antibody to the molecule and so it may not be the best choice for the first generation of antibodies developed.

**HasAP.** Aerobic metabolism of *P. aeruginosa* requires iron for many of the respiratory enzymatic processes. HasAp is a secreted hemophore that, after being cleaved by a *Pseudomonas* protease [176], captures iron as part of the heme acquisition system [176]. Its release is regulated via *quorum sensing* (QS) in response to cell growth and biofilm establishment [177]. HasAp is not constitutively expressed, but of the QS-regulated proteins studied, iron regulation proteins are the most abundant [178]. Targeting protease activity, or the hemophore activity of HasAp, could help eliminate chronic infection by starving *P. aeruginosa* of essential iron needed for aerobic metabolism, and thus may be a worthy therapeutic antibody target.

### 3.8. Immunomodulators

Immunomodulators use regulatory molecules to adjust the immune response, including inducing, attenuating, and amplifying the host immune response. *P. aeruginosa* uses several mechanisms to modulate the immune system to reduce effective host immune response, but *P. aeruginosa* also upregulates the immune response to fight competitors such as *S. aureus*. For example, *P. aeruginosa* increases IL-8 production and in response, *S. aureus* dampens Toll-like receptor (TLR1/TLR2)-mediated activation of the NF-κB pathway [179]. As noted earlier, *P. aeruginosa* uses LasA to inhibit competing *S. aureus*. Here, a few immunomodulators are highlighted as targets during a *P. aeruginosa* infection.

**Lipoxygenase (LoxA).** Lipoxygenases (LOXs) are extracellular, lipid-oxidizing enzymes that have potent immunoregulatory properties in their hosts [180,181]. In *P. aeruginosa*, the lipoxygenase LoxA oxidizes polyunsaturated fats, resulting in the expression of bioactive lipid mediators and suppression of chemotactic molecules [182]. With greater than 90% conservation across *P. aeruginosa* strains, leukotrienes, a product of LoxA, are secreted to balance lipoxins and resolvins. LoxA also activates host ferroptosis, resulting in host cell death and inflammation. LoxA increases LXA4 along with other LOX expressions, leading to an increase in neutrophil chemotaxis and pro-inflammatory signaling [183]. Evidence from preclinical animal studies shows that *P. aeruginosa* clinical isolates secrete LoxA, indicating modulation of the host immune response during acute pneumonia [182]. However, LoxA decreases in expression after 24 h of lung infection and has decreased expression in chronic infections. This indicates that LoxA is a significant pro-inflammatory and bacterial invasion mediator in acute infections, while losing this effect when lung infections become chronic. Thus, targeting LoxA during acute infection could help increase the host immune response. Additionally, upregulation of CCR5, a receptor on white blood cells for immune system chemokines, occurs after exposure of lung tissues to Lox through unknown mechanisms. CCR5 acts as a chemokine receptor as well as a chemotactic agent, resulting in increased bacterial invasion [182]. Since LoxA is accessible to both circulating and secreted (IgA and IgM) antibodies, it may prove to be a good target for therapeutic antibody development and delivery therapy.

**Leukocidin.** Leukocidin is a cytotoxin of *P. aeruginosa* that damages granulocytes and lymphocytes by forming a beta-barrel pore in cell membranes, thereby inhibiting essential bactericidal and phagocytic immune functions [184,185]. Leukocidin has long been known as a common, secreted virulence factor in *P. aeruginosa.* Targeting Leukocidin, which is accessible, ubiquitous, and pathogenic, could have direct influences on the course of *P. aeruginosa* infection and could help increase the host immune response.

## 4. Antibodies as Therapeutics

As antibiotic resistance continues to increase throughout a wide spectrum of microbial infections, new and innovative approaches to treating these infections must be sought. The development of antibodies to the relevant bacterial targets outlined above and in Table 1 may be the next step in combating antimicrobial resistance. Therapeutic antibodies work by activating and modulating our own host effector mechanisms including direct neutralization of toxins and pathogens, activation of the complement pathway, activation of neutrophil and macrophage opsonophagocytosis, activation of natural killer cells, enhancement of antigen presentation from dendritic cells to T cells and follicular dendritic cells to B cells, and degranulation of mast cells, eosinophils, and basophils. Additionally, Fc receptors that correspond to different classes of antibodies can activate the complement pathway, induce innate immune responses, and enhance natural adaptive immunity, providing a multi-faceted strategy to address disease [186].

Therapeutic monoclonal antibodies (mAbs) have emerged as a tour de force in the drug market and are projected to capture a rising percentage of the worldwide drug market—a trend that reflects increased prescription use of existing therapeutic mAbs and a large number (~50) of new therapeutic mAb drugs approved for use [187]. Currently, mAbs have been developed to treat a broad array of clinical conditions including cancer, autoimmune diseases, transplants, infectious disease, and toxin neutralization [188].

In contrast to mAbs, polyclonal antibodies (pAbs) have multiple epitope binding sites, thereby allowing greater coverage of neutralizing antigens. Much like our native immune system, pAbs are produced as a collection of antibodies from multiple B cell lineages, thus producing many different sites and affinities for the same antigen. Having multiple epitope binding sites allows for greater neutralization options and less opportunities for pathogens to develop escape mutants [189]. By harboring the ability to work at a wide number of sites in microbial pathogenesis, polyclonal antibodies have the potential to curb or slow the development of multidrug-resistant organisms and fill the gaps in treatment left by antibiotics and vaccines. Passive polyclonal antibodies have been used to treat diseases for over 100 years with the oldest being patient- and blood-derived products (often “convalescent sera”) or intravenous immunoglobulins (IVIG) [190]. Additionally, more modern pAbs are currently used as antitoxins and antivenoms and are an effective treatment post-infection or post-exposure for botulism, varicella, vaccinia, rabies, CMV, hepatitis B, and tetanus [188]. This is not a new strategy, but merely a re-energized approach to treating bacterial pathogenesis.

In an emerging appreciation of rational design for new therapeutic options, combinations of mAbs with defined correlates of protection are driving an increasing number of investigations into combination therapies, especially with a focus on broad effector mechanisms of actions beyond just simple binding. This strategy has been useful in recent viral outbreaks, where multiclonal cocktails, or convalescent sera (i.e., polyclonal) from recovered patients, have been used to treat diseases [189]. Engineering combination therapies with improved effector functions, specificity, affinity, and glycosylation generates a stronger and more diverse immune response. Moreover, combination therapies also generate a much higher barrier for pathogens to overcome; in other words, the pathogens cannot easily evolve to circumvent the treatment [189].

### Combinatorial Therapy

Recently, it has been shown that antibodies and antibiotics can act concomitantly or even synergistically against pathogens. Several studies have shown that exposure to antibiotic therapy alters the expression of secreted factors, cell surface proteins, and other cellular products that may be viable immunotargets [190]. These altered levels of expression can change and potentially enhance the innate and humoral immune responses to pathogens—namely through increased antigen exposure, increased protein secretion, and potentially decreased virulence of pathogens [190]. Inspired by these same principles, combination or conjugate antibiotic/antibody therapeutic strategies, particularly those based on polyclonal antibodies, could be designed to capitalize on these same altered expression levels to enhance therapeutic efficacy and combat resistance. Below is a non-exhaustive look into the evidence demonstrating how combination antibiotic/antibody therapy can have a synergistic effect and, in some cases, potentiate the antimicrobial response.

Several clinical isolates of *Streptococcus pneumoniae* have been shown to, when in the presence of penicillins or cephalosporins, become more reactive to opsonization by the innate immune response; acute phase reactants, such as c-reactive protein and serum amyloid P, more efficiently bind cellular antigens and clear infection quicker than when not in the presence of these antibiotics [190]. By examining the synergistic activity between antimicrobial drugs and innate immune responses in vivo, it can by deduced that combinations between antimicrobial and antibody therapy will likely work in a similar synergistic fashion.

Similar to making *Streptococcus pneumoniae* more readily opsonized, penicillins, cephalosporins, and gentamicin have all been shown to alter gene expression and transcription in *Staphylococcus aureus*, leading to a change in the molecular structure of the cell walls of the organisms [191]. Surprisingly, changes in the expression of virulence factors in *S. aureus* in response to antibiotic therapy may actually improve exposure to cellular products targeted by polyclonal antibody therapies. Using this observation as a foundation, further work is underway to characterize the response of organisms to specific antimicrobial therapies, allowing polyclonal antibodies to be tailored to the expected upregulation of virulence factors. In 2015, Lehar et al. utilized rifamycin conjugated to monoclonal antibodies against wall-teichoic acids (cell wall peptidoglycans) to eliminate residual methicillin-resistant *S. aureus* infections [191]. These antibiotic–antibody conjugates effectively bound *S. aureus* cell walls allowing rifamycin to be taken up by phagolysosomes and exert its antimicrobial effect intracellularly, showing improved eradication rates compared to Vancomycin treatment alone [192]. Such an effect was confirmed in a recent in vivo study.

A monoclonal antibody to PcrV (in *P. aeruginosa*) was used in combination with either ciprofloxacin, tobramycin, or ceftazidime to improve survival rates in mice, noting that the effect was more pronounced in more virulent strains of *P. aeruginosa* [192]. Protecting the lung against the cytotoxic effects of exotoxin in addition to amplifying the antimicrobial effects of the antibiotics highlights the value of antibiotic–antibody conjugate therapy and punctuates the need for further research into the human applications of these treatments.

## 5. Discussion

Therapeutic antibodies are highly specific with long half-lives and minimal dosing compared to other drug treatments [193]. Currently, there are 79 US FDA-approved monoclonal antibodies in use and well over 500 currently being studied in clinical trials [193]. Most therapeutic antibodies target cancer, autoimmune diseases, or prevent transplant rejection, but their use in other disease realms is gaining momentum. Currently, there are five approved infectious disease monoclonals: Bezlotoxumab for prevention of recurrent *Clostridiodes difficile*, Ibalizumab for drug-resistant HIV, Palivizumab for prevention of respiratory syncytial virus, and Raxibacumab and Obiltoxaximab for inhalational anthrax, and casirivamab and imdevimab for COVID-19 [194].

New therapies and treatment options must be sought for *P. aeruginosa*, a highly prevalent and exceedingly antibiotic-resistant microorganism. Patients with the highest risk of infection include those with compromised immunity, seen with chronic burns, chronic wounds, and HIV/AIDS patients. Those with deficient bacterial clearance mechanisms, as seen in cystic fibrosis, COPD, and ventilator-dependent patients [195], are also particularly susceptible to *P. aeruginosa*. *P. aeruginosa’s* widely varied environmental niches, from respiratory tract infections, skin, and soft tissue infections, and even septicemia make treatment especially challenging. Additionally, *P. aeruginosa* creates seemingly impenetrable biofilms, which are especially prevalent on implanted and indwelling devices, such as catheters, endotracheal tubes, and prostheses. These biofilms create a plethora of challenges for the host innate and adaptive immunity as well as antimicrobials [92]. The most daunting challenge, however, may be the ability of *P. aeruginosa* to modulate host immune responses.

As noted earlier, multidrug-resistant pathogens present an emergent, serious public health threat at all levels. While we have detailed many of the potential targets of *P. aeruginosa*, many other emerging infections are multipathogenic, multidrug-resistant, and even more deadly. With poly- and oligoclonal mixtures of therapeutic antibodies, increasing levels of efficacy can be reached by adjusting individual antibodies within the mixture for specific or broad pathogenic threats. The oligomixture can also have prophylactic applications with adjustable dosing. Further, oligomixtures and antibiotics could be used concomitantly and potentially synergistically against individual or multiple pathogens. While vaccination is a good option, oligomixtures have significant advantages and offer a strong alternative to treat multidrug resistance pathogens. Most notable is the ability to neutralize identified, virulent targets without the inherent variability in response to vaccination.

The effector function of an antibody plays a critical role in pathogen neutralization. With an oligomixture, the role of effector function can be engineered (see Figure 5). This is especially true when compared to a vaccine in which effector function is merely a function of genetic and/or environmental chance. With oligomixtures, the effector function options could be optimized based on any number of factors, including genetics. Effector functions could be standardized for the entire mixture or diversified to allow different parts of the immune system to enter the fight. The isotypes and thus the effector functions of the oligomixture could be highly dependent on the environment. For example, IgA could be the most efficacious isotype for intestinal and endothelial pathogenic indications whereas IgG isotypes would likely be most efficacious in sepsis indications.

Another major consideration moving forward will be drug delivery. As detailed in the body of this review, the common indications that will benefit from polyclonal therapies are found in diverse environments. In most cases detailed in this manuscript, the traditional therapeutic concentrations of antibiotics in serum and at the site of infection are below MIC levels, decreasing the efficacy of the treatment and increasing the probability of antibiotic resistance. Ideally, the route would be dependent on the site of infection. For bacteremic infections, intravenous would be the route of choice. Beyond that, it gets trickier. A potential route would be topical administration of IgA/IgG via a nebulizer solution for pneumonia or patients with cystic fibrosis. Studies have supported this possibility; IgG and IgA that have been nebulized in a variety of solutions and viscosities have been shown to increase immunoglobulin levels in bronchoalveolar lavage specimens up to 48 h after administration as well as confer immunity to the specific immunoglobulin targets [196]. Potential for topical preparations for keratitis, folliculitis, and other soft tissue infections exist as well, expanding the range of these therapeutics [197]. Recently it has been shown that antibodies and antibiotics can act concomitantly or even synergistically against pathogens. In particular, bacterial cell surfaces can be altered by antibodies, resulting in multiple mechanisms that improve drug efficacy [198,199].

*P. aeruginosa*, like many other bacteria, continues to and will continue to find ways to subvert the efficacy of antibiotics. The immune system has evolved to attack microbes on multiple fronts, but *P. aeruginosa* likewise evolves counterattacks, allowing it to colonize and infect hosts. Engineering combination antibody therapies with numerous targets as presented here would be the best option currently available to improve therapeutic efficacy and reduce the opportunity for resistance development. As the *P. aeruginosa* genome is characterized or evolves, the combination of antibodies can be adapted and optimized. Most importantly, with the improved engineering of antibodies and characterization of the human genome, the engineering of more efficacious antibody technologies for diverse indications can also evolve, or even become personalized, just as *P. aeruginosa* evolves.

## Figures and Tables

**Figure 1 antibiotics-10-01530-f001:**
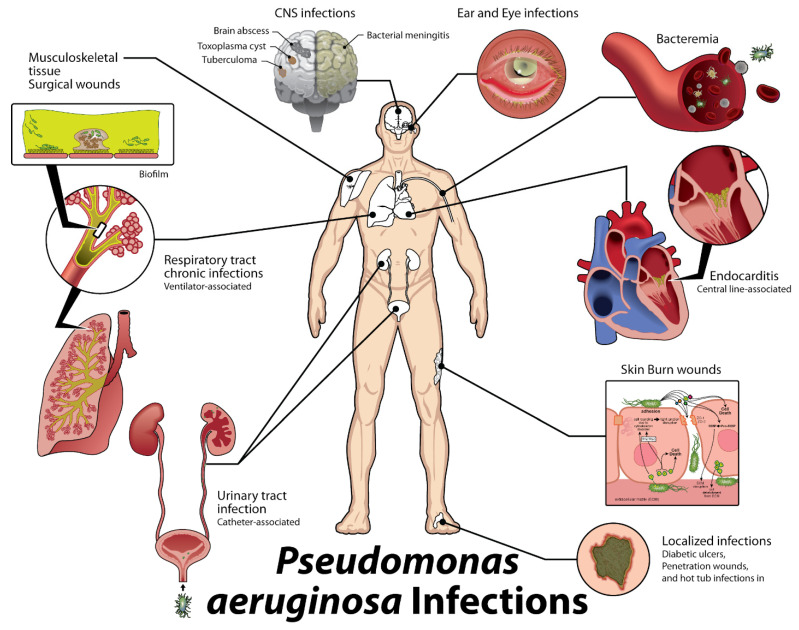
Types of Acute *P. Aeruginosa* Infections [5]. *P. aeruginosa* is prevalent in skin and soft tissue infections (top right) including trauma, burns, and dermatitis. It also commonly causes swimmer’s’ ear (external otitis), hot tub folliculitis, and ocular infections, bacteremia and septicemia, especially in immunocompromised patients, and endocarditis associated with IV drug users and prosthetic heart valves (bottom right). *P. aeruginosa* can also cause central nervous system (CNS) infections such as meningitis and brain abscess (top left), bone and joint infections, including osteomyelitis and osteochondritis, respiratory tract infections, and hospital-acquired urinary tract infections (UTIs; bottom left). *P. aeruginosa* is also resistant to many common antibiotics [5].

**Figure 2 antibiotics-10-01530-f002:**
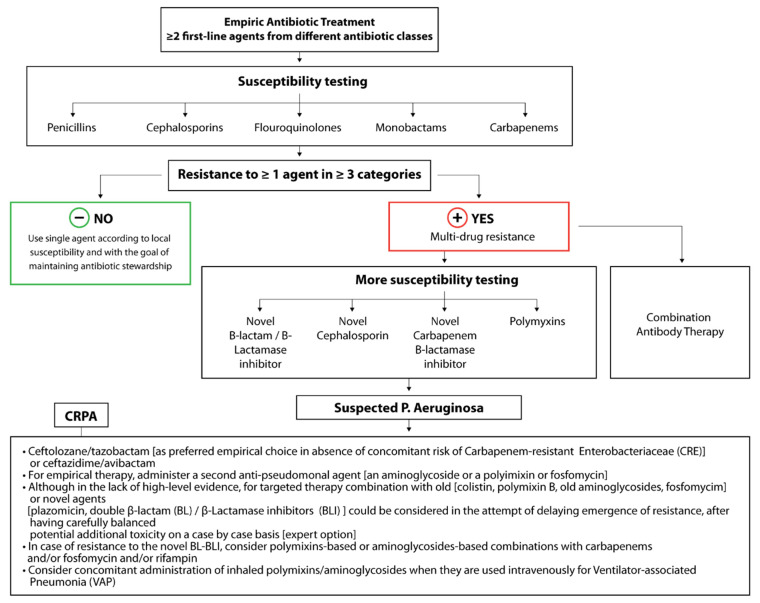
Treatment strategy for carbapenem-resistant *P. aeruginosa* isolates including future treatment options based on combinatorial antibody therapies [21].

**Figure 3 antibiotics-10-01530-f003:**
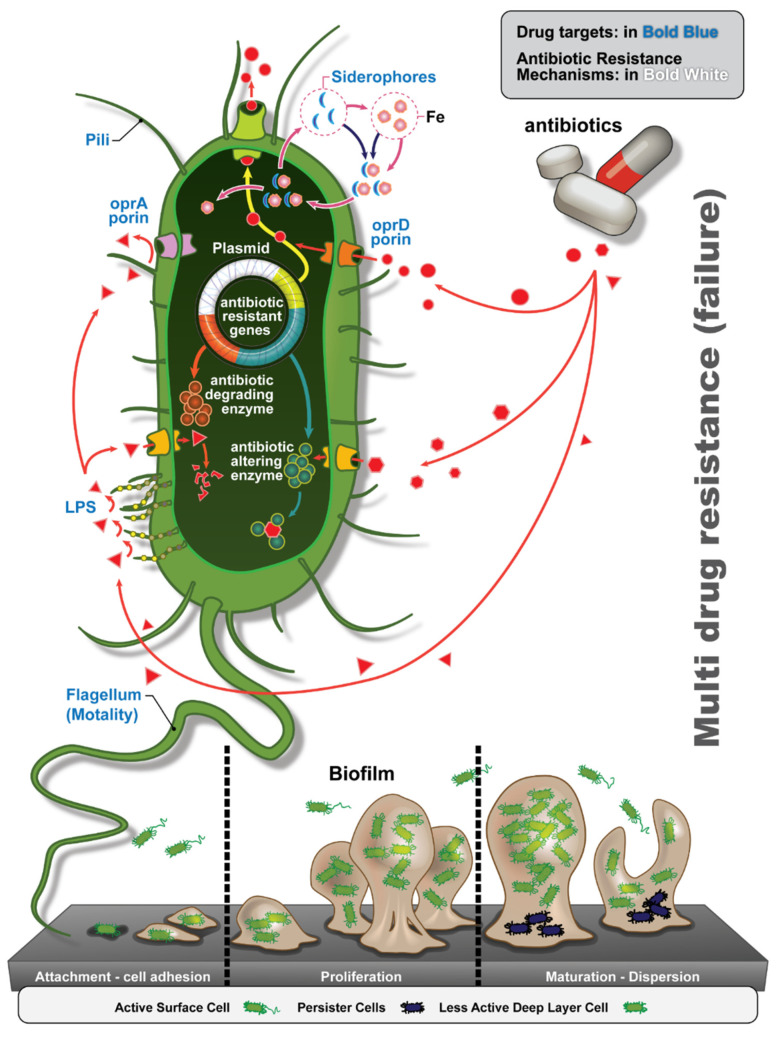
Mechanisms of antibiotic resistance in *P. aeruginosa*. These include all of the mechanisms in blue and biofilms.

**Figure 4 antibiotics-10-01530-f004:**
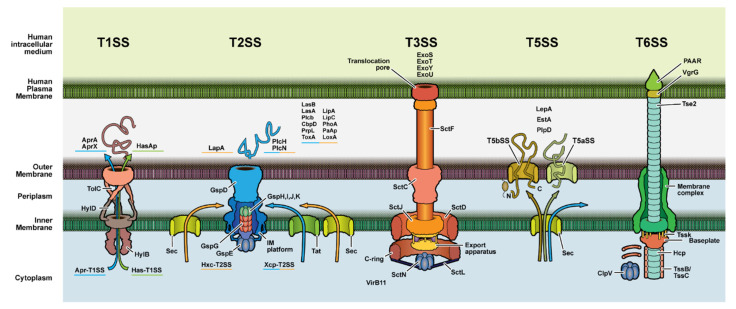
Protein secretion systems in *P. aeruginosa* described further in the text.

**Figure 5 antibiotics-10-01530-f005:**
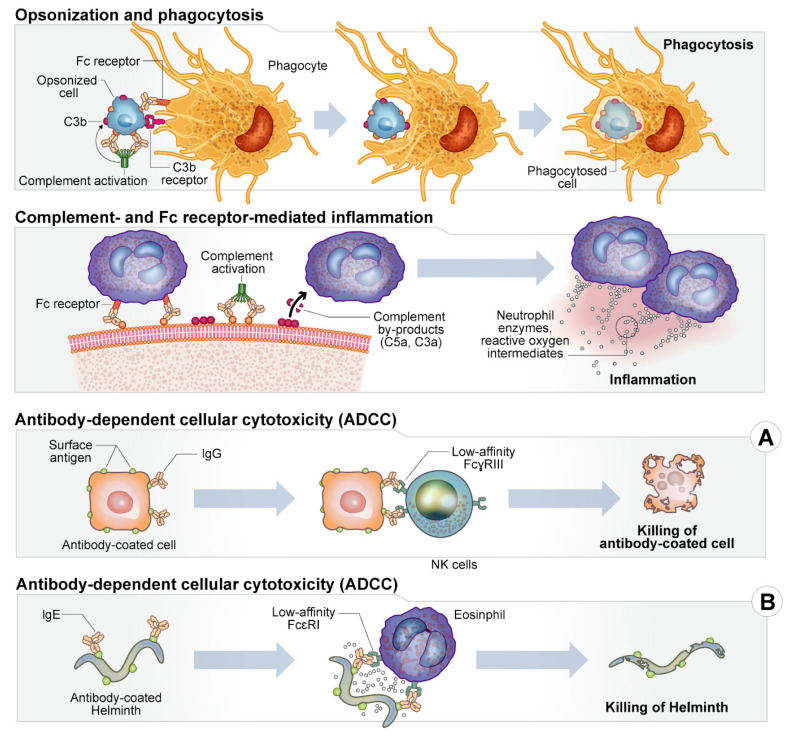
Effector function or mechanisms of killing by antibodies.

**Table 1 antibiotics-10-01530-t001:** Potential Therapeutic Antibody Targets.

Location or Class	Examples	Activity/Effects on Host
Cell surface	Alginate	Antiphagocytic, resists opsonic killing
	Lipopolysaccharide	Endotoxic, antiphagocytic, avoids preformed antibody to previously encountered O antigens
	Pili (produced by type IV secretion)	Twitching motility, biofilm formation, adherence to host tissues
	Flagella	Motility, biofilm formation, adherence to host tissues and mucin components
	Injection of type III secretion factors	PcrG, PcrV, PcrH, PopB, and PopD proteins form injection bridge for type III effectors
Outer membrane	Siderophore receptors	Provides iron for microbial growth and survival
	Efflux pumps	Remove antibiotics
Secretion systems		
Type II	Elastase, lipase, phospholipases, chitin-binding protein, exotoxin A, and others	Variety of proteolytic, lipolytic, and toxic factors; degrade host immune effectors
Type III	ExoS, ExoT, ExoU, ExoY	Intoxicates cells (ExoS, ExoT); cytotoxic (ExoU); disrupts actin cytoskeleton
Type VI	Cytoplasmic and membrane-associated proteins, ATPases, lipoproteins, Hcp1 protein	Poorly characterized but found in animal studies to be needed for optimal virulence, particularly in chronic infection
Iron acquisition	Pyoverdin, pyochelin, HasAP	Scavenge iron from the host for bacterial use
Secreted toxins	Hemolysins, rhamnolipid phospholipases	Kill leukocytes, hemolysis of red cells, degrade host cell surface glycolipids
Secreted oxidative factors	Pyocyanin, ferric pyochelin, HCN	Produce reactive oxygen species: H_2_O_2_, O_2_^−^ Inflammatory, disrupts epithelial cell function
*Quorum sensing*	LasR/LasI, RhlR/RhlI, PQS	Biofilm formation, regulation of virulence factor secretion

ATPases = adenosine triphosphatases; PQS = *Pseudomonas* quinolone signal.

## Data Availability

Data sharing not applicable.

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
