# Peer review of "Potential Therapeutic Targets for Combination Antibody Therapy against Pseudomonas aeruginosa Infections"

_antibiotics, 2021, doi:10.3390/antibiotics10121530_

Round 1

Reviewer 1 Report

Proctor et al. provide a review and update about potential targets for antibody treatment against Pseudomonas aeruginosa infections. The article is timely and especially welcome in concern of the emerging multi-resistant strains of P. aeruginosa. After the general introduction into this topic, the authors specifically describe and propagate multiple therapies (i) with combinations of monoclonal antibodies against various bacterial targets or (ii) immunotherapies combined with the known antibiotics. In chapter 1, a more detailed description about the (metabolic?) factors driving the switch from planktonic forms into the more pathogenic biofilm forms could be interesting and merits more attention. I am also suggesting to give more details about the mechanism involved in the formation of persister cells. Chapter 2 summarizes the host immune responses and their limitations. In chapter 3, appropriate targets for immunotherapies are described in quite detail including secreted toxins and invasins, proteins involved in the machineries of secretion systems, factors and metabolites involved in quorum sensing, triggers determining antibiotic resistance, appropriate membrane components, factors involved in motility, siderophores, and other immunomodulators. This core part represents a most detailed and updated compilation of factors that could be exploited in future for immunotherapies against Pseudomonas. In the next chapter, the authors briefly describe some aspects of immunotherapies and finalize their review with a well balanced discussion again highlighting the concepts of future immunotherapies against P. aeruginosa infections.

The text is carefully written and explains all relevant aspects with very nice text-book quality illustrations. Together, the article is not only a welcome update and outlook for experts in the field, but also a comprehensive and interesting article for the general readership of the Journal. The manuscript should be accepted for publication.

Author Response

Thank you very much for your detailed feedback.

Reviewer 2 Report

In the manuscript ID: antibiotics-1474372 by Proctor and colleagues, the authors present a detailed review of potential targets for antibody therapies against the opportunistic pathogen Pseudomonas aeruginosa. After underlining the pathogenic potential of the microorganism and the caused infectious processes, they list all the potential antibody targets, describing their functions and reporting the published studies about the related immunological therapy.

The paper is interesting and sounding and provides detailed information about the anti-Pseudomonas immunological therapy, which is still quite neglected in comparison to antibiotic therapy.

There are, however, some concerns to be addressed before publication:

-As section 4 “Antibodies as Therapeutics” describes the general action of antibodies in therapy, it should be placed as general section, before listing the potential P. aeruginosa targets for the development of specific antibodies;

 -For some potential targets there is no indication of published studies about their immunological role, while the target itself is extensively described for its involvement in the bacterial cell physiology and pathogenicity. Considering the general focus of the review on the antibody therapy, the authors should consider reducing the related sections, especially number 3.5, 3.7 and 3.8, providing just a briefer description of the proposed targets;

-Although cited (lines 907-910), there is too little explanation about the combinatory antibodies/antibiotics therapies, which on the contrary would deserve a dedicated section to describe the results obtained so far;

-Please provide a better-quality Figure 2 and substitute the heading Table 2 with Table 1.

After these major revisions, the manuscript can be considered eligible for publication in “Antibiotics”.

MINOR COMMENTS

-Please type “quorum sensing” in Italic throughout the manuscript;

-Please after the first citation of “Pseudomonas aeruginosa”, use “P. aeruginosa” throughout the manuscript;

-Please substitute “percent” with “%” throughout the manuscript;

-Please correct “Gram-negative” throughout the manuscript;

-Lines 30-33, please rephrase the sentence to make it clearer;

-Line 66, please substitute “generalist” with “a common environmental pathogen”;

-Lines 144 and 717, please type “P. aeruginosa” in italic;

-Line 161, please delete the question mark after “consuming”;

-Line 239, please substitute “Hemolytic” with “The hemolytic one”;

-Lines 285-287, please explain the sentence;

-Line 405, please type “in vitro” in italic;

-Line 464, please correct “It acts as a quorum sensor signal molecule”;

-Line 598, please type “ompA” in italic;

-Line 718, please delete the point after “genetic material”;

-Line 749, please correct “making it an attractive candidate” with “makes it an attractive candidate”.

Author Response

Thank you very much for your detailed feedback. We appreciate your time.

Reviewer 3 Report

 The authors widely covered recent trends in combination antibody therapy against Pseudomonas aeruginosa infections. They addressed current hurdles to combat infections. In addition, mechanisms of antibiotic resistance, current treatment strategy against antibiotics resistance and combination therapy against infections were also covered in this manuscript.   This review is relevant in the field. The authors widely discussed current scenarios and future directions of therapy against P.  aeruginosa infections.

Tables and figures are good, but incorrectly cited in the main texts.

Line 716: Figure X? Is it Figure 4?

Line 809: There is no Table 1. Please include it.

Please refer to the attached file for comments. 

Author Response

(The authors gave the same response as above.)

Round 2

Reviewer 2 Report

In the revised version of the manuscript ID: antibiotics-1474372 by Proctor and colleagues, the authors have successfully addressed the main raised concern, i.e. the lack of a section related to antibodies-antibiotics combinatory therapies. As the other critics can be considered stylistic divergences, the paper can now be published in “Antibiotics”.